# Transcriptome-Wide Identification and Expression Analysis of *bHLH* Family Genes in *Iris domestica* under Drought and Cu Stress

**DOI:** 10.3390/ijms25031773

**Published:** 2024-02-01

**Authors:** Qiang Ai, Mei Han, Cuijing Liu, Limin Yang

**Affiliations:** College of Chinese Medicinal Materials, Jilin Agricultural University, Changchun 130118, China

**Keywords:** *Iris domestica*, *bHLH* transcription factor, expression pattern, isoflavones

## Abstract

The role of *bHLH* transcription factors in plant response to abiotic stress and regulation of flavonoid metabolism is well documented. However, to date, the *bHLH* transcription factor family in *Iris domestica* remains unreported, impeding further research on flavonoid metabolism in this plant. To address this knowledge gap, we employed bioinformatics to identify 39 *IdbHLH* genes and characterised their phylogenetic relationships and gene expression patterns under both drought and copper stress conditions. Our evolutionary tree analysis classified the 39 *IdbHLHs* into 17 subfamilies. Expression pattern analysis revealed that different *IdbHLH* transcription factors had distinct expression trends in various organs, suggesting that they might be involved in diverse biological processes. We found that *IdbHLH36* was highly expressed in all organs (Transcripts Per Million (TPM) > 10), while only 12 *IdbHLH* genes in the rhizome and four in the root were significantly upregulated under drought stress. Of these, four genes (*IdbHLH05*, -*37*, -*38*, -*39*) were co-upregulated in both the rhizome and root, indicating their potential role in drought resistance. With regards to copper stress, we found that only 12 genes were upregulated. Further co-expression analysis revealed that most *bHLH* genes were significantly correlated with key enzyme genes involved in isoflavone biosynthesis. Thereinto, *IdbHLH06* showed a significant positive correlation with *IdC4H1* and *Id4CL1* (*p* < 0.05). Furthermore, a transient expression assay confirmed that the IdbHLH06 protein was localised in the nucleus. Our findings provide new insights into the molecular basis and regulatory mechanisms of *bHLH* transcription factors in isoflavone biosynthesis in *I. domestica*.

## 1. Introduction

*I*. *domestica* is a perennial herbaceous medicinal plant in the Iridaceae family, which is widely cultivated in various regions of China such as Jilin, Liaoning, Henan, Hubei, and Yunnan [1,2,3]. The plant is renowned for its high content of isoflavones, which exhibit a range of pharmacological activities including anti-inflammatory, antibacterial, antiviral, antioxidant, and anticancer effects [4]. In China, rhizomes are the traditional medicinal site of *I*. *domestica*, but isoflavones are also present in the roots, stems, and fruits, with varying levels among different parts [3,5], which could be due to environmental factors and gene expression [6,7]. So far, the isoflavone biosynthesis pathway in *I. domestica* has not been fully elucidated; only key enzyme genes have been cloned such as *PAL1*, *C4H*, *4CL*, *CHS*, *CHI1*, and *CHI2* and believed to be involved in regulating isoflavone biosynthesis [5]. These genes are functionally conserved, and their expression patterns which in turn affect isoflavone biosynthesis are regulated by specific transcription factors [8].

The *bHLH* transcription factor family is a significant group of transcription factors with the basic region of amino acids containing leucine residues and helix-loop-helix domain [9]. Typically composed of around 60 to 80 amino acids, these proteins contain two structural domains: the N-terminal basic region and the C-terminal HLH region [10]. The basic region generally comprises polar amino acids, including leucine, arginine, and lysine, which are capable of forming stable complexes with the negatively charged DNA backbone. The HLH region consists of two helices, which can form dimers and interact with other proteins to regulate gene transcription [11]. Notably, in the biosynthesis of flavonoids, *bHLH* transcription factors interact with MYB transcription factors to co-regulate the expression of key genes involved in flavonoid biosynthesis. This interaction affects traits such as flower colour, fruit colour, and disease resistance [12,13]. In recent years, a growing body of research has highlighted the crucial role of the *bHLH* transcription factor family in regulating the diversity and specificity of plant secondary metabolite synthesis [14]. And further investigation of the *bHLH* transcription factor family’s role in plant secondary metabolite synthesis will reveal the biosynthetic mechanisms underlying these compounds and provide theoretical and practical guidance for developing medicinal plants with high quality and yield.

Research has demonstrated that *bHLH* transcription factor family members mainly utilise two mechanisms of action. Firstly, they act as direct transcription factors that bind to the E-box on the target gene promoter, thereby regulating the gene’s transcription level [15]. Secondly, they can serve as transcription factor co-activators, interacting with other transcription factors such as MYB, bZIP, and WRKY to co-regulate gene expression [16,17]. In plant secondary metabolite biosynthesis, the bHLH transcription factor family typically acts as a co-activator of MYB transcription factors, co-regulating the expression of key genes involved in flavonoid biosynthesis. For instance, in *Arabidopsis*, MYB75 and bHLH2 function as co-regulators of gene expression in the flavonoid biosynthesis pathway [18]. The MYB75/bHLH2 complex binds to the promoter region of the phenylalanine ammonialyase (*PAL*) gene, regulating its expression, and subsequently affecting the biosynthesis of flavonoids [19]. Furthermore, studies have demonstrated that *bHLH* transcription factor family members can also regulate gene expression in biological processes such as anthocyanin biosynthesis, pollen development, and disease resistance, exerting significant effects on plant growth, development, and environmental adaptation [20]. Therefore, thoroughly investigating the mechanism and regulatory network of plant *bHLH* transcription factors in secondary metabolite biosynthesis will provide new ideas and technical support for the breeding and production of medicinal plants. Additionally, it will have an important driving effect on the study of plant biology and gene regulatory mechanisms.

This article aims to conduct a comprehensive analysis and study of the *bHLH* transcription factor family in *I. domestica*. Firstly, we will use bioinformatic methods to classify and identify the *bHLH* transcription factor family in *I. domestica* and conduct analyses on their basic structure and phylogenetics. Secondly, we will examine their spatiotemporal expression patterns in different organs under drought and Cu stress and calculate the Pearson’s correlation of candidate isoflavone biosynthesis genes and *IdbHLH* genes. Finally, we will verify the subcellular localization of the candidate gene *IdbHLH06* using laser confocal microscopy. Our findings will provide new insights into the roles of *IdbHLHs* in regulating isoflavone biosynthesis in *I. domestica* and contribute to the research on plant biology and gene regulatory mechanisms.

## 2. Results

### 2.1. Identification of bHLH Gene Family in I. domestica

A total of 108 *bHLH* genes were obtained through predictions from the NR, Swiss-Prot, KEGG, and COG databases. The online tool ORF Finder was used for online prediction of *IdbHLH* sequences, of which 45 genes had complete ORF (open reading frame). After removing four redundant sequences, 39 *bHLH* genes (designated as *IdbHLH01*-*IdbHLH39*) were retained for subsequent analysis. The corresponding gene ID, gene length, protein length, CDS (Coding sequence) length, MW (molecular weight), PI (Isoelectric point), instability index, aliphatic index, and grand average of hydropathicity were listed in Appendix A. The gene length of these sequences ranged from 390 to 1671 bp, with an encoding range of 130 to 557 amino acids. The MW of these proteins ranged from 14.88 to 61.03 kDa, and their pI varied from 4.62 to 12.17. Analysis of instability index, aliphatic index, and grand average of hydropathicity suggested that these proteins were hydrophilic and unstable. Subcellular localization analyses revealed that most *IdbHLH* proteins were located in the nucleus, with a few found in chloroplasts (*IdbHLH05*, *-06*, *-07*, *-18*, *-19*, *-31*, and *-33*, Appendix A).

### 2.2. Phylogeny and Multiple Sequence Alignment of IdbHLH Protein

In order to investigate the phylogenetic relationships among *IdbHLHs*, we constructed an unrooted phylogenetic tree using the maximum likelihood method. The analysis included a total of 197 *bHLH* proteins, consisting of 39 *IdbHLHs* and 158 *AtbHLHs*, and the resulting tree is presented in Figure 1. The unrooted tree demonstrated that the *IdbHLHs* were divided into 17 subfamilies, named from I to XVII, with 3 *bHLHs* not classified into any family as orphans, including 1 *IdbHLH42* and 2 *AtbHLHs* (*AtbHLH007* and *AtbHLH059*). Notably, each clade did not contain independent *bHLH* protein except the orphan, and the number of *IdbHLH* members in each clade varied widely from 1 (clade III) to 6 (clade VII and XII). These findings suggested that there is a similar pattern of individual evolution and functional diversity between *I. domestica* and *A. thaliana*.

### 2.3. Conserved Motif Analysis of IdbHLH Protein

To investigate the structural features of *IdbHLH* proteins, we analysed the conserved motifs in these proteins using MEME software (Figure 2). Proteins linked closely on the evolutionary tree had similar motif constitutions. Interestingly, different subfamilies had distinct motif patterns, indicating that *bHLHs* in the same subfamily may have a similar function. Motif 1, 2, and 4 were shared by 33, 28, and 13 subfamily genes, respectively, while motif 6, 7, 8, 9, and 10 were present in only 2 subfamily genes. Furthermore, the number of motifs in each subfamily gene varied widely, ranging from 1 to 6. Among them, 8 genes had only 1 motif, while 2 genes had up to 6 motifs.

### 2.4. Expression Profiles of IdbHLH in Different Organs

The expression levels of *IdbHLH* in different plant organs, including the root, rhizome, aerial stem, leaf, flower, and fruit, were determined using TPM values derived from publicly available RNA-seq data [21,22]. A heatmap of *bHLH* gene expression was generated (Figure 3), revealing that only four genes (*IdbHLH22*, -*27*, -*32*, and -*34*; TPM = 0) were not expressed in any tissue. Of the remaining 35 genes (89.7%), 26 (66.7%) were expressed in all organs, while the expression patterns of the remaining genes were categorized into low expression, tissue-specific expression, and high expression. For instance, *IdbHLH01~03* genes showed low expression levels, with expression values less than 1. Two genes, *IdbHLH08* and *IdbHLH10*, exhibited tissue-specific expression, with the former being highly expressed only in rhizomes (TPM = 11.22), and the latter being highly expressed in the root (TPM = 9.40), rhizome (TPM = 12.07), and aerial stem (TPM = 8.01). The gene with high expression levels in all organs was *IdbHLH36* (TPM > 10). These results indicate that the expression patterns of *IdbHLH* transcription factors may be involved in various biological processes in diverse organs.

### 2.5. Expression Profiles of IdbHLHs in Response to Drought and Copper Stress

We analysed the expression pattern of *IdbHLHs* in the rhizome and root under drought stress using our transcriptome data, and in the root under copper stress using shared transcriptome data (Figure 4 and Figure 5). Overall, the transcription factors in the rhizome and root were expressed under all treatments (TPM > 0), but there were clear differences in their expression patterns. Under drought stress, only 12 *IdbHLH* genes in the rhizome and 4 in the root were greatly upregulated. Four of these genes (*IdbHLH05*, -*37*, -*38*, -*39*) were co-upregulated in both organs, suggesting that they may have an important role in drought resistance. Additionally, 6 genes (15.4%) in the rhizome were continuously downregulated, and over half of the *IdbHLH* genes in the root (21 genes, 53.8).

### 2.6. Co-Expression of Candidate Isoflavone Biosynthesis Genes and IdbHLH Genes

*bHLHs* play a role in the production of plant secondary metabolites such as tanshinones by activating key enzyme genes in the biosynthetic pathway [23]. We conducted a co-expression analysis of *IdbHLHs* and candidate isoflavone biosynthesis genes to explore the association between *IdbHLHs* and isoflavone biosynthesis (Figure 6 and Figure 7). The analysis revealed that the expression patterns of *IdbHLHs* could be categorised into three clusters. Cluster I comprised of most *IdbHLHs* that were highly correlated with *IdPAL1*, with *IdbHLH29* and -*33* showing the strongest correlation. In Cluster II, most *IdbHLH* genes were strongly correlated with *IdC4H1*, *Id4CL1*, and *IdCHI*. For instance, *IdbHLH06* showed significant positive correlation with *IdC4H1* and *Id4CL1* (*p* < 0.05). In Cluster III, *IdbHLH* genes were only positively associated with *IdC4H1* and *Id4CL1*, but negatively related to other isoflavone biosynthesis genes.

### 2.7. Subcellular Localization

To elucidate the potential function of *IdbHLH06* and verify its subcellular localization, the 16318h-*IdbHLH115*-GFP recombinant was transformed into *Arabidopsis* protoplasts using the PEG4000-mediated method, and the fluorescence signal of GFP was observed through laser confocal microscopy. The results revealed that IdbHLH115 proteins were localized in the nucleus (Figure 8, Bar = 10 μm), which is consistent with previous predictions. As with many other transcription factors, IdbHLH115 may play a role in the transcriptional regulatory system.

## 3. Discussion

The *bHLH* transcription factor family is a crucial family of transcription factors present widely across eukaryotes, and plays a role in several growth and developmental processes and biosynthetic pathways in plants [10]. In medicinal plants, *bHLH* transcription factors regulate the biosynthesis pathway of secondary metabolites [9]. Therefore, investigating the effects of *bHLH* transcription factors on secondary metabolites in medicinal plants holds significant theoretical and practical value. In this study, we report the systematic analysis of the *bHLH* transcription factor family in *I. domestica*, based on our transcriptome data and public data. Additionally, we analysed the phylogenetic relationships, conserved motifs, and expression profiles of *IdbHLH* genes under drought and copper stress. This study is the first to identify and report on the *bHLH* transcription factor family in *I. domestica*.

At the outset, a total of 108 *IdbHLH* genes were identified, which is fewer than the *bHLH* genes found in *Arabidopsis* (162 genes) and maize (167 genes) [24]. Nonetheless, this is also fewer than the 182 *bHLH* genes identified from six *I. domestica* organs, based on transcriptome data [5,25]. This difference in the number of genes could be due to variations in sequencing depth or screening thresholds. Ultimately, only 39 *IdbHLH* genes were selected for further analysis. The phylogenetic analysis indicated that *I. domestica* and *Arabidopsis bHLH* genes (197) belonged to 17 subfamilies, excluding orphans. However, as shown in 638 *bHLH* genes from *Arabidopsis thaliana*, *Populus trichocarpa*, *Oryza sativa*, *Physcomitrella patens*, and 5 algae species, the number of *bHLH* gene subfamilies may increase with the expansion of species and gene numbers, increasing to 32 subfamilies [26]. The protein motif analysis revealed that most members of each group shared similar motifs and positions, which is a typical feature observed in other plants [27]. This suggests that proteins of the same subfamily have similar evolutionary origins and molecular functions.

The biosynthesis of plant secondary metabolites involves complex pathways, which can be activated by abiotic stress-induced transcription factors that promote the expression of key enzyme genes [28]. *bHLH*, one such transcription factor family, regulates flavonoid biosynthesis pathways, such as the GA regulatory network of baicalin [29,30] and the regulation of tanshinone [31]. While previous studies have shown that drought and Cu stress can increase the content of isoflavones in *I. domestica* [3,32], it remains unclear how *bHLH* transcription factors regulate isoflavones in this plant. In this study, the expression of 39 *IdbHLH* transcription factor genes and six key enzyme genes was investigated under drought and Cu stress, and their Pearson correlation was evaluated. Consistent with previous studies, many *bHLH* genes were found to be involved in plant responses to drought and Cu stress, such as *ZmPTF1* in maize, which positively regulates *NCED* target genes to participate in drought response [33], and *bHLH38*, whose overexpression can increase Cu absorption [34]. There were also significant differences in the expression levels of each *bHLH* gene among different organs [10,35], with nearly 50% of *bHLH* expression levels found to be high in different organs, including the root hairs under Cu stress. Under sustained drought conditions, the expression of most *bHLH* genes was elevated in the roots and rhizomes. However, some *bHLH* genes responded differently to abiotic stress, such as *IdbHLH06*, which was upregulated under drought stress but not under Cu stress (Figure 4 and Figure 5). Co-expression analysis of *IdbHLH* and key enzyme genes revealed a positive correlation between 4/5 *IdbHLH* (16 genes) and *IdC4H1* and *Id4CL* (*p* < 0.05, Figure 7). Additionally, Hou demonstrated that *bHLH42* was involved in activating *C4H* and *4CL* to regulate anthocyanin accumulation in sweet potato storage roots [36]. Based on these findings, a molecular mechanism network for abiotic stress-*bHLH*-isoflavone biosynthesis was constructed, in which drought and Cu regulate key enzyme genes by affecting the expression of *IdbHLH*, thereby influencing the biosynthesis of isoflavones (Figure 9). The study also verified the subcellular localization of IdbHLH06 protein, which was found to be located in the nucleus and may function by regulating structural genes. The results provide a foundation for further research into the biosynthesis mechanism of isoflavones in *I. domestica*.

## 4. Materials and Methods

### 4.1. Plant Materials and Drought Treatment

*I. domestica* seeds from our previous research were cultivated artificially at the Medicinal Plant Cultivation and Physiological and Ecological Practice Teaching Base [3]. *I. domestica* plants (2 years old) were selected and cultivated in a rain-proof shed (43°48′ N, 125°25′ E) at Jilin Agricultural University under natural photoperiod (approximately 12 h of light), with an average humidity of 67.81% and an average temperature of 24.5 °C from April to August 2020. Each plant was given 150 mL of tap water daily to support normal growth before the onset of drought treatment. On 1 September, watering was suspended, and the drought treatment was initiated, allowing soil moisture to drop naturally, and the change of soil water content was detected every day.

Following the classification of drought grades based on the soil relative humidity index (R-value) [37] and considering the historical August rainfall of 130 mm in Changchun, China, over the past 50 years as a reference, root and rhizome samples were collected under varying drought stress conditions. These conditions included no drought stress (25.0% soil water content, CK), mild-moderate drought stress (18.0% soil water content, MLW), moderate-severe drought stress (~14.3% soil water content, MDW), and severe drought stress (~2.0% soil water content, SVW), with 3 biological replicates for each treatment. The sampling protocol was in accordance with our prior study on root physiological changes and transcriptional analysis of *I*. *domestica* in response to prolonged drought [3]. All collected samples were rapidly frozen in liquid nitrogen and stored at −80 °C.

### 4.2. Identification of bHLH Protein in I. domestica

The Hidden Markov Model file for the *bHLH* domain (PF00010) was obtained from Pfam version 35 (http://pfam.xfam.org/, accessed on 3 July 2023) and used to search for previous transcriptome data of *I. domestica* with the HMMER software version 3.3.2 (http://hmmer.org/, accessed on 3 July 2023). The NCBI-Conserved Domain Database version 3.19 (CCD, accessed on 3 July 2023) was then used to validate all candidate genes under default parameters. Only unique genes with complete CDS were selected for further analysis, including protein sequence acquisition with NCBI ORF Finder (https://www.ncbi.nlm.nih.gov/orffinder/, accessed on 3 July 2023) and physical and chemical parameters such as amino acids, MW, and pI using ExPASy ProtParam (http://www.expasy.org/, accessed on 3 July 2023). Subcellular localization was predicted using WoLFPSORT (https://wolfpsort.hgc.jp/, accessed on 3 July 2023).

### 4.3. Multiple Sequence Alignment, Conserved Motif, and Phylogenetic Analyses

The *bHLH* protein sequences of *Arabidopsis thaliana* were obtained from the plant-TFDB database version 2.0 (accessed on 3 July 2023). Multiple sequence alignment was performed using ClustalX2 software version 2.1 (accessed on 3 July 2023). The IQ-TREE program version 2.2.2.6 (accessed on 3 July 2023) was utilized with the parameters of JTT+F+R5 model, 1000 ultrafast bootstrap approximation, and 1000 SH-aLRT to build a phylogenetic tree. The phylogenetic tree was further annotated using iTOL version 6 (https://itol.embl.de/, accessed on 3 July 2023). To discover motifs in sets of *IdbHLH* sequences, MEME program version 5.4.1 (accessed on 3 July 2023) was used with the default parameters, except for a maximum of 10 motifs and an optimum motif width of 6–200 amino acid residues. The motifs results was visualized using TBtools software version 1.098769 (accessed on 3 July 2023) [38].

### 4.4. Expression Patterns Analysis of IdbHLH Genes

Two public SRA datasets of *I. domestica* were retrieved from NCBI’s website (https://www.ncbi.nlm.nih.gov/sra, accessed on 3 July 2023) comprising various organs (root, rhizome, aerial stem, leaf, flower, and fruit) from three-year-old plants (PRJNA596865) [5], and roots of one-year-old plants treated with CuCl_2_ for 0 and 24 h (PRJNA596865) [39]. For analysing the expression of *IdbHLH* genes under drought, same samples as in Section 4.1. were utilized for building a transcriptome library and sequencing via an Illumina NovaSeq 6000 platform (LC Sciences, Houston, TX, USA). The Trinity software version 2.4.0 (accessed on 3 July 2023) was used for de novo assembly of the transcriptome. The CDS of the *IdbHLH* gene served as the search sequence (query). Utilizing this query sequence, a search for similar sequences was conducted in various transcriptome databases (accession number: PRJNA430284) with a threshold of e < 10^−5^. Subsequently, the expression levels of the *IdbHLH* gene across root, rhizome, aerial stem, leaf, flower, and fruit were computed using Salmon software version 1.9.0 (accessed on 3 July 2023). The expression amounts were normalised using the TPM method. HeatMap plugin version 1.098769 (accessed on 3 July 2023) was used to draw expression heatmap [38].

### 4.5. Subcellular Localization Analysis

Gene cloning: Total RNA samples (LC Science, Houston, TX, USA) were extracted using the Total RNA Purification Kit, and cDNA synthesis was conducted with the Reverse Transcription Kit (Takara, San Jose, CA, USA). The cDNA corresponding to the *IdbHLH06* gene was amplified via RT-PCR using specific primers (Appendix A). Subsequent PCR amplification employed the first-strand cDNA as a template, following the system and amplification conditions outlined by Ai [40]. The resulting PCR product was integrated into *pEASY*-T1 vectors, introduced into competent cells (TransGen, Beijing, China), and the positive cloning solutions were subjected to sequencing and identification (Comate Bioscience Co., Ltd., Changchun, China).

Subcellular Localization Vector Construction: (1) Target gene PCR amplification—Double enzyme cleavage sites were designed based on the multiple cloning site (MCS) of the 16318h-GFP vector. *Pst* I and *Nco* I restriction enzymes were employed for the *IdbHLH06* gene, and primers with homologous recombination arms (Appendix A) were crafted. The cDNA fragments were amplified using the *IdbHLH06* gene cloning vector plasmid as a template. The reaction products were recovered, purified through agarose gel electrophoresis for subsequent use. The amplification system and conditions were the same as above. (2) Vector linearization—The 16318h-GFP vector underwent digestion at 37 °C for 1 h and 65 °C for 20 min. The agarose gel was then recovered, and the purified vector was reserved for later use. (3) Ligation of the target fragment to the expression vector—Employing seamless cloning technology, the target fragment was ligated with the linearized vector using the ClonExpress^®^ II One Step Cloning Kit (Vazyme Biotech Co., Ltd., Nanjing, China, C112). The ligation system comprised 1 μL of linearized vector, 3 μL of DNA, 4 μL of 5 × CE II Buffer, 2 μL of Exnase II, and 10 μL of d_2_H_2_O. (4) Recombinant plasmid transformation and gene validation—The transformed recombinant plasmid was introduced into colibacillus DH5α competent cells. PCR identification of the bacterial solution was performed (Appendix A, PCR reaction system as mentioned above), and the preliminary screening of positive recombinants was conducted. The remaining bacterial solution was preserved in glycerol at a concentration of 50%, and the rest was sequenced for verification (Comate Bioscience Co., Ltd.).

Finally, the recombinant plasmids were transfected into *Arabidopsis* protoplasts via PEG-mediated gene transfer, and the alterations in fluorescence were observed using laser confocal microscopy (LEICA SP8, Wetzlar, Germany) [24].

## 5. Conclusions

This study presents the first comprehensive and systematic analysis of the *bHLH* gene superfamily in *I. domestica*. We identified 39 *IdbHLH* genes and characterised their motif structure and evolutionary conservation. Based on phylogenetic comparisons with *Arabidopsis*, these genes were classified into 17 subfamilies. Additionally, a transient expression assay confirmed the nuclear localisation of the IdbHLH06 protein, indicating its probable regulation of structural genes. Interestingly, *IdbHLHs* were found to respond to both drought and Cu stress, suggesting their involvement in the biosynthesis of isoflavones in *I. domestica*. Further experiments, such as genetic function verification, will be conducted. Taken together, our findings provide insight into the molecular basis and regulatory mechanisms of *bHLH* transcription factors in the biosynthesis of isoflavones in *I. domestica*.

## Figures and Tables

**Figure 1 ijms-25-01773-f001:**
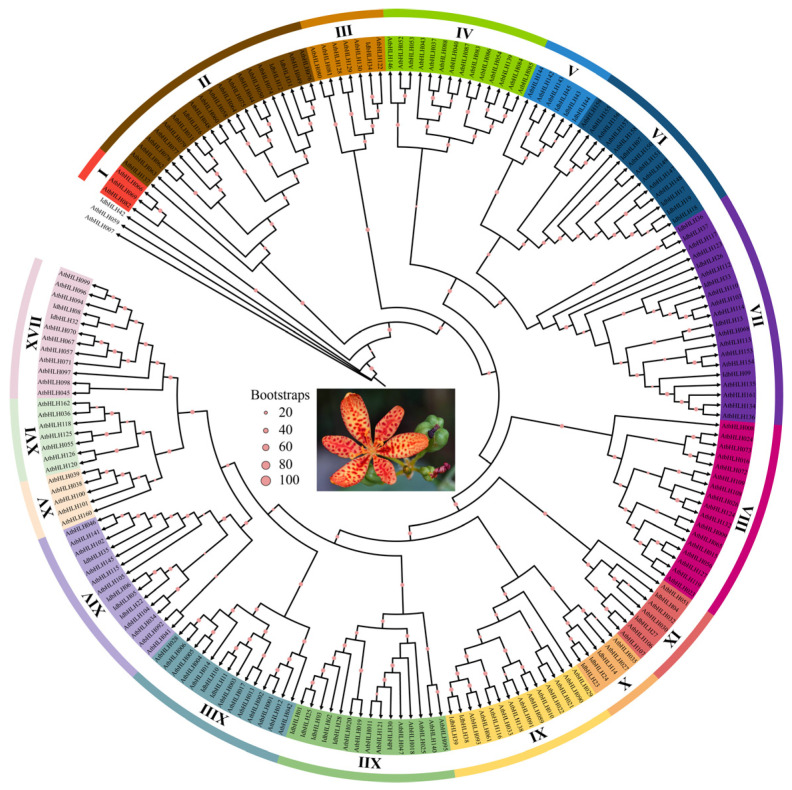
The construction of a phylogenetic tree for *bHLH* proteins in *Arabidopsis* and *I. domestica*. The Maximum Likelihood tree was constructed utilizing the JTT+F+R5 model parameters, along with 1000 ultrafast bootstrap approximations and 1000 SH-aLRT tests.

**Figure 2 ijms-25-01773-f002:**
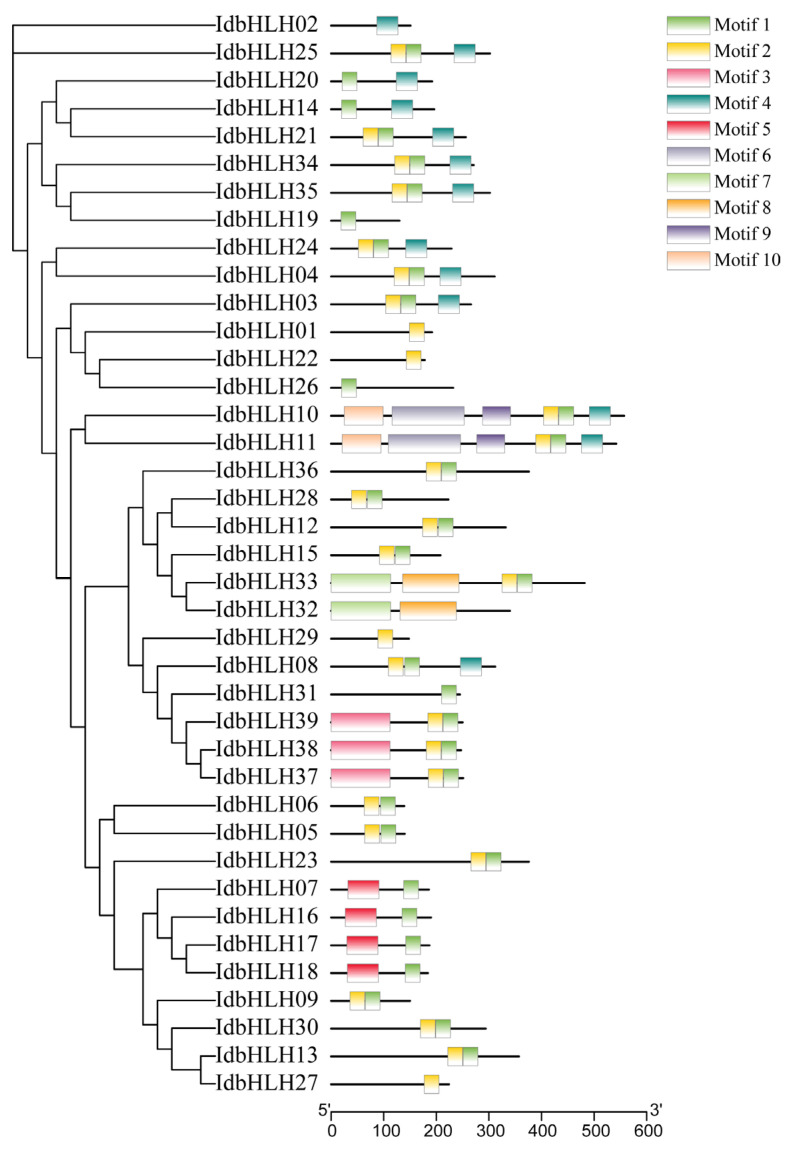
Analysis of conserved motifs of *IdbHLH* members. Ten predicted motifs are displayed in different coloured boxes.

**Figure 3 ijms-25-01773-f003:**
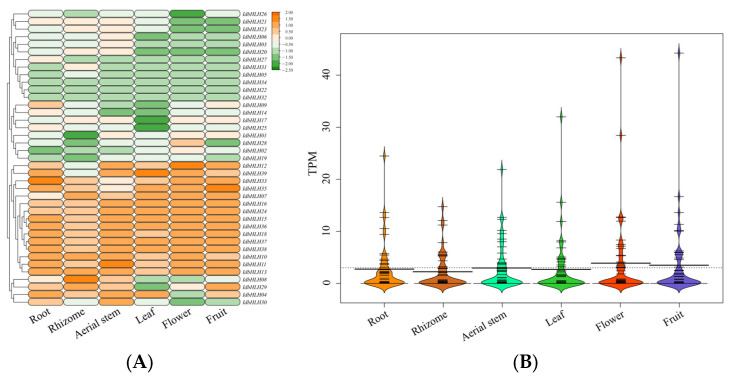
Expression heatmap (**A**) and beanplot (**B**) of *IdbHLH* gene across various organs (root, rhizome, aerial stem, leaf, flower, fruit). In subfigure (**B**), the *y*-axis represents the TPM value and the *x*-axis represents the different groupings. The thick black horizontal lines represents the median TPM for each group, while the dotted line represents the median of all data. The following image is the same as this one.

**Figure 4 ijms-25-01773-f004:**
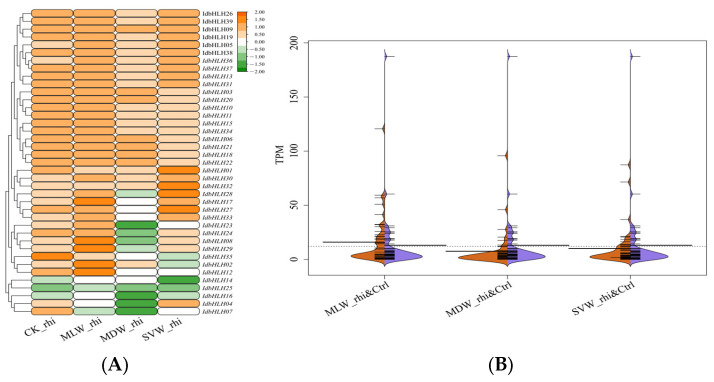
Expression heatmap (**A**) and beanplot (**B**) of *IdbHLH* gene in rhizome under different drought conditions.

**Figure 5 ijms-25-01773-f005:**
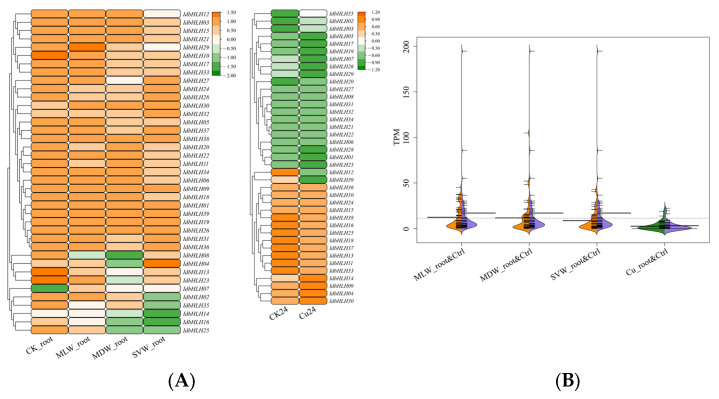
Expression heatmap (**A**) and beanplot (**B**) of *IdbHLH* gene in root under different drought conditions and Cu stress.

**Figure 6 ijms-25-01773-f006:**
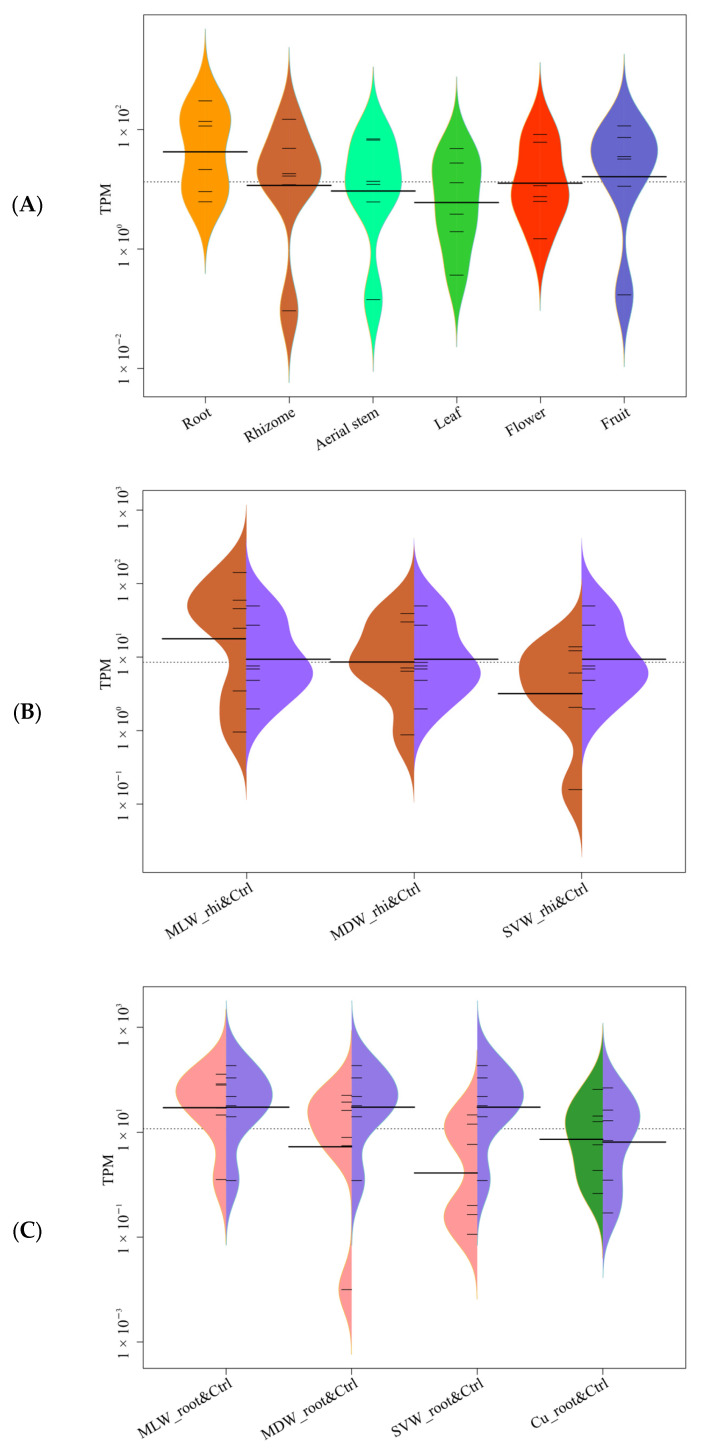
Expression beanplot of six key enzyme genes across various organs (**A**), in rhizome (**B**) and root under different drought conditions and Cu stress (**C**).

**Figure 7 ijms-25-01773-f007:**
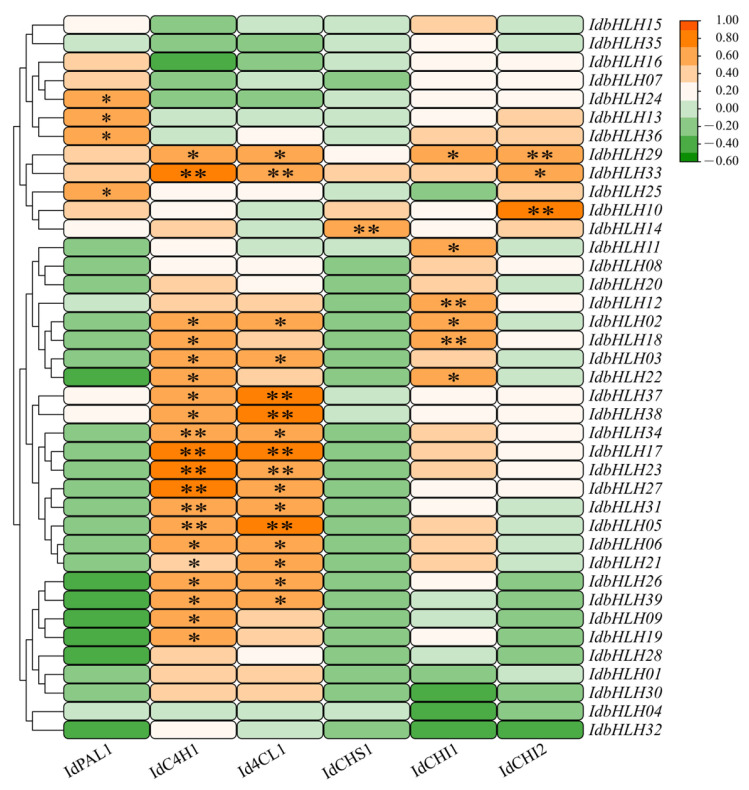
The Pearson’s correlation coefficients between *IdbHLHs* and six key enzyme genes. *PAL*, phenylalanine ammonia lyase; *C4H*, cinnamate 4-hydroxylase; *4CL*, p-coumaroyl coenzyme A ligase; *CHS*, chalcone synthase; *CHI*, chalcone isomerase. The “*” denotes statistical significance at the 0.05 level, determined through Duncan’s Single-factor variance analysis. The “**” indicates significance at the 0.01 level, as determined by Duncan’s Single-factor variance analysis.

**Figure 8 ijms-25-01773-f008:**
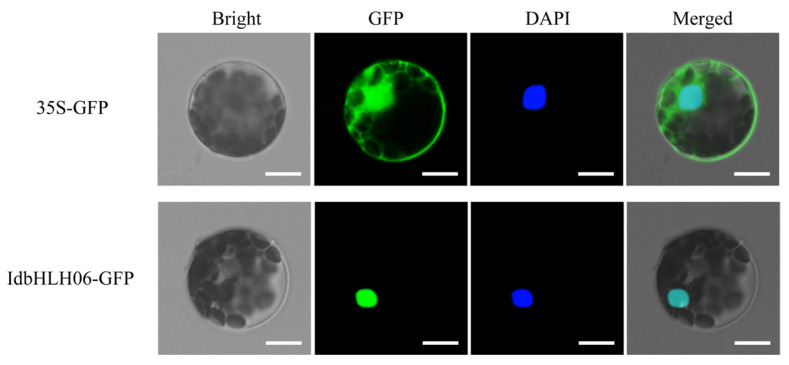
Subcellular localization of IdbHLH06 protein. Transient expression of 35S-GFP and 16318h-*IdbHLH06*-GFP recombinant plasmids in *Arabidopsis* protoplasts by PEG-mediated gene transfer; the GFP signal was observed by laser confocal microscopy, Bar = 10 μm.

**Figure 9 ijms-25-01773-f009:**
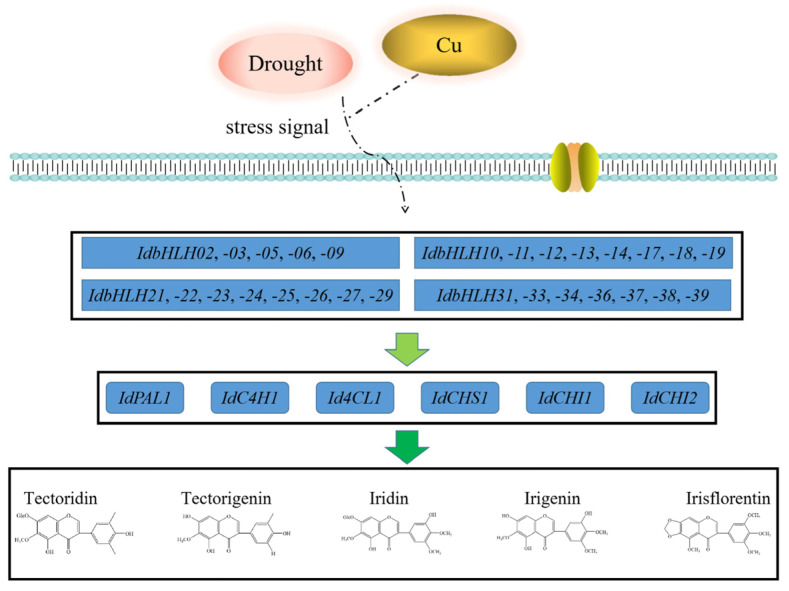
A molecular mechanism network for abiotic stress-*bHLH*-isoflavone biosynthesis. Drought and Cu stress regulate key enzyme genes by affecting the expression of *IdbHLH*, thereby influencing the biosynthesis of isoflavones. The dotted line indicates that the path is not clear.

## Data Availability

Data is contained within the article and Appendix A.

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
