# Peer review of "Transcriptome-Wide Identification and Expression Analysis of bHLH Family Genes in Iris domestica under Drought and Cu Stress"

_ijms, 2024, doi:10.3390/ijms25031773_

Round 1

Reviewer 1 Report

Comments and Suggestions for Authors

This article presents a study on bHLH transcription factors in Iris domestica, focusing on their role in plant response to abiotic stress and regulation of flavonoid metabolism. The study identified 39 IdbHLH genes, characterized their phylogenetic relationships, and investigated their expression patterns under drought and copper stress conditions.

 The analysis of the published data was provided with a sufficient level of scientific novelty. The text of the paper is concrete, realistic, and understandable. However, there are important flaws in the manuscript listed below:

 - In the entire manuscript, all scientific names should be italicized.

 - The title of the manuscript needs to be changed according to the experiments. No practical experiments demonstrate bHLH involvement in isoflavone biosynthesis. Additionally, there should be more conclusive results mentioned in the conclusion.

 - In the Materials and Methods (lines 100-102), some methods need to be mentioned in detail. For example, why were these days used for collecting the samples? How was the drought condition initiated? Also, why was the sampling different from the sampling used for gene expression (lines 128-131)? The differences and the reasons for these samplings should be mentioned clearly.

 - In the Materials and Methods, the expression profiles of IdbHLH genes were determined in various tissues. However, it is unclear what was used for this evaluation and how the results were considered for the next step of cloning and generating transgenic expression.

 -The study mainly focused on bioinformatics and expression pattern analysis and additionally requires experimental validation (excluding transgenic expression) of the functional roles of the identified IdbHLH genes. Therefore, in the conclusion section, further experiments for more validation of the results should be recommended.

Comments on the Quality of English Language

Minor editing of English language required

Reviewer 2 Report

Comments and Suggestions for Authors

Dear Authors,

I have read the manuscript "Transcriptome-wide analysis of bHLH transcription factors involved in regulating isoflavone biosynthesis in Iris domestica“. The focus of the study is to conduct a comprehensive analysis and study of the bHLH transcription factor family in I. domestica. The analysis should be deepened from the practical side to identify the function of the bHLH genes.

The Bioinformatics methods are original. It is a great amount of work with scientific relevance. The gap is the experimental work which needs to be deepened by investigating the expression patterns of the genes under drought and copper stress conditions.

The introduction needs to be revised. In some places in the text, it is very difficult to understand what the authors mean. Also, some numbers appear in the text.

The method, It's a novel study, because bHLH in I. domestica remains unknown to date. However, The authors need to explain why they chose the specific time points for the treatments. The drought experiment needs to be revealed in detail. For the copper treatment, 24 hours is too short a period, in my opinion. It needs to be extended with more time points or generally removed from the manuscript. Like this doesn't give any sufficient information at all.

In the conclusion, the authors need to indicate what further experiments they have planned to continue the work. Also, regarding the drought and copper stress, these analyses are preliminary and general, and specific conclusions should not be drawn.

The references are appropriate.

The additional comments:

Figures 3, 4 and 5 need to be replaced with new one, because they are out of focus and unreadable.

The quality of English language needs to be improved, by minor editing

I recommend major revision.

Comments on the Quality of English Language

The quality of English language needs to be improved by minor editing.

Round 2

Reviewer 1 Report

Comments and Suggestions for Authors

-

Author Response

Thank you for giving us the opportunity to submit a revised draft of the manuscript“Transcriptome-wide Identification and Expression Analysis of bHLH Family Genes in Iris domestica under Drought and Cu Stress”for publication in the Journal of International Journal of Molecular Sciences. We appreciate the time and effort that you and the reviewers dedicated to providing feedback on our manuscript and are grateful for the insightful comments on and valuable improvements to our paper.

We have incorporated most of the suggestions made by the reviewers. Those changes are highlighted within the manuscript. Please see below, in red, for a point-by-point response to the reviewers’ comments and concerns. All page numbers refer to the revised manuscript file with tracked changes. Because of your suggestions, the revised articles become better and readers can get more valuable information. Thanks again to the editors and reviewers for their help.

Thank you for your review of the manuscript and we will continue to refine this paper.

Kind regards,

Qiang Ai

[email protected]

Reviewer 2 Report

Comments and Suggestions for Authors

Dear Authors,

The manuscript has been improved and the quality of English language is better. The Figures are with good quality. Please, chek the title for a missing letter in „drought“. Also, the names of the genes and latin names of the plants shoul be in Italic. There is some omissions there. In  the conclusion you removed the information regarding drought and copper stress in general. My suggestion was to avoid general conclusions because the analyses are only in silico and preliminary, and they are not supported by molecular analysis through RT-PCR, for example. But this doesnt mean to delete the conclusions at all, just to revice them.

Due to the slight corrections in the text that need to be made, I recommend minor revisions. After this, the manuscript will be suitable for publication.
